# Do the Determinants of Mental Wellbeing Vary by Housing Tenure Status? Secondary Analysis of a 2017 Cross-Sectional Residents Survey in Cornwall, South West England

**DOI:** 10.3390/ijerph19073816

**Published:** 2022-03-23

**Authors:** Richard A. Sharpe, Katrina M. Wyatt, Andrew James Williams

**Affiliations:** 1European Centre for Environment and Human Health, University of Exeter Medical School, Truro TR1 3HD, UK; richard.sharpe@cornwall.gov.uk; 2Wellbeing and Public Health Service, Cornwall Council, Truro TR1 3AY, UK; 3Relational Health Group, University of Exeter College of Medicine and Health, Exeter EX1 2LU, UK; k.m.wyatt@exeter.ac.uk; 4Population and Behavioural Science, School of Medicine, University of St Andrews, St Andrews KY16 9TF, UK

**Keywords:** adult, mental health, housing, tenure, prevention, community, risk factors, protective factors, living conditions, neighbourhood circumstances

## Abstract

Housing is a social determinant of health, comprising multiple interrelated attributes; the current study was developed to examine whether differences in mental wellbeing across housing tenure types might relate to individual, living, or neighbourhood circumstances. To achieve this aim, an exploratory cross-sectional analysis was conducted using secondary data from a county-wide resident survey undertaken by Cornwall Council in 2017. The survey included questions about individual, living, or neighbourhood circumstances, as well as mental wellbeing (Short Warwick-Edinburgh Mental Wellbeing Scale). A random sample of 30,152 households in Cornwall were sent the survey, from whom 11,247 valid responses were received (38% response), but only 4085 (13.5%) provided complete data for this study. Stratified stepwise models were estimated to generate hypotheses about inequalities in mental wellbeing related to housing tenure. Health, life satisfaction, and social connectedness were found to be universal determinants of mental wellbeing, whereas issues related to living circumstances (quality of housing, fuel poverty) were only found to be related to wellbeing among residents of privately owned and rented properties. Sense of safety and belonging (neighbourhood circumstances) were also found to be related to wellbeing, which together suggests that whole system place-based home and people/community-centred approaches are needed to reduce inequalities.

## 1. Introduction

Globally, mental health disorders are the greatest contributors to years lived with disability in young adults and the second highest contributors in adults aged between 50 and 69 years [1]. Mental wellbeing is influenced by diverse social determinants of health (e.g., various social, economic, and physical risk factors) and social inequalities experienced from before birth throughout one’s life course [2,3]. Housing is widely recognised as a social determinant of health, with wellbeing influenced by the standard of housing (living circumstances) as well as the perceptions and experiences of living in that neighbourhood and ability to access services (neighbourhood circumstances) to support living and ageing well [4,5,6,7]. It is important to understand the wider health risks and benefits associated with a household’s social structure, ecology of place or surrounding neighbourhoods [8], as well as the natural environment, culture, and history of the community [4,5,6,7]. These diverse and complex relationships with others and with the local environment can create the conditions for improved health and wellbeing [9]. Including the place and community—as part of the system by which health outcomes, such as good and bad mental wellbeing, are generated and often reproduced over the life course, as well as transmitted from generation to generation—is therefore a necessary consideration to improve the health of the public [4,10,11].

Housing tenure status, such as owning your own home or living in rented accommodation, has been found to be a factor influencing mental health and wellbeing. The tenure under which someone resides in a property will be related to their individual, living, and neighbourhood circumstances. Therefore, although housing tenure has been identified as a determinant of mental wellbeing, the specific experiences and circumstances that contribute to the reported inequality in wellbeing, especially the interactions among individual, living, and neighbourhood circumstances, are less well understood. We address this gap in the literature through secondary analysis of data from a survey of residents from Cornwall in the United Kingdom. The use of this survey permitted the concurrent estimation of the contributions of individual, living, and neighbourhood circumstances to the wellbeing of residents of privately owned, privately rented, and social housing properties.

## 2. Summary of Current Research

Inequalities in mental wellbeing by housing tenure status have been reported in several countries [12,13,14]. In the UK, universal health benefits, such as lower levels of ill-health, anxiety, and depression, are often assumed to be found in those owning their home when compared to those living in the rental sector [15]. In Canada, the UK, and Austria, it has been found that those living in the private rental sector have higher levels of distress, especially when compared with those owning their own home; with those owning their own home without a mortgage having the lowest distress levels [12,16]. Despite there being some protection of rental rights in Germany, those living in this rental sector still report poorer self-rated health when compared to homeowners [17]. In the United States of America, Fenelon et al. [18] found that, compared to those waiting for federal housing assistance, those in receipt of housing assistance in the form of public or multifamily housing (similar to social housing in the UK) reported better health and wellbeing, and this association was not mediated by neighbourhood circumstances. Lawder et al. [19] found associations between neighbourhood housing tenure mix and wellbeing related health outcomes in Scotland, with lower self-reported health and higher alcohol-related illness in areas with more social housing.

A systematic literature review by Alidoust and Huang [7] summarised the associations between housing and health into the following four topics: neighbourhood or context, physical building, housing market, and housing policy. McElroy et al. [20] found individual, community, and place characteristics, which together contributed to wellbeing in a network analysis in the UK. Li et al. [13] documented the detrimental impacts of instability in the private rental sector on mental wellbeing using longitudinal data in Australia. Pollack et al. [17] found that the association between housing tenure and health in Germany was mediated by need for home renovations (living circumstances), perceptions of noise or air pollution, and relationships with neighbours (neighbourhood circumstances) [17]. Tonn et al. [14] also found that the association between poverty and health was not linear in relation to housing, with those living in mobile homes or multifamily buildings reporting less hardships than those in single-family homes. In the UK, lower income households living in the social housing sector can experience higher levels of home renovations (e.g., energy efficiency improvements) when compared to the private rental and ownership sector due to fiscal interventions across the social housing sector and the need to comply with the Decent Homes Standard [21]. Home ownership may enable higher income households to make choices about where they live, improve home equity (the amount of money tied up in the home), and other individual and societal benefits [4]. These may include improved neighbourhood circumstances, such as living in stable community environments and reduced crime [22]. However, these higher income households can be adversely affected by affordability and unemployment or financing and rising mortgage interest rates, which can affect one’s mental wellbeing [23,24]. Furthermore, these potential negative experiences may be concealed by the size of the home ownership sector where there may be short spells of poverty, which depend on variable socio-demographic and spatial differences [25]. This body of research highlights that there are multiple intersecting factors related to individual, living, and neighbourhood circumstances that contribute to housing tenure-based inequalities in wellbeing.

This complexity is further demonstrated by the interventions that have been implemented in relation to housing. Despite a number of interventions targeting living conditions, many households experience persistent problems with the indoor environment, which influence the health and wellbeing of residents [26,27,28,29]. Alongside the living conditions within a home (e.g., temperature, damp, mould), diverse interrelated factors, such as the component of ‘place’, and many socioeconomic factors, such as housing affordability and tenure status, contribute to housing-related health outcomes [30,31]. ‘Place’ is important because those living in very low income areas can feel poorly served by local services, regardless of the tenure of their housing, this can then lead to a feelings of abandonment and isolation from other people living in the neighbourhood and, hence, poor health and wellbeing [9]. Consequently, there have been questions about the extent to which the private and social rental sectors should provide homes, not just houses [32,33]. Garnham et al. [32] (p. 1) found that, in Glasgow, for renters, a home was a ‘recuperative space in which to shelter from daily stressors and was a source of autonomy and social status’, which was underpinned by the ‘housing service, property quality and affordability’. Consequently, living in a safe, well maintained, and affordable home, as well as social connections with others in the community and employment, are key contributors to mental wellbeing. Our aim was to examine whether the differences in mental wellbeing across housing tenure types might relate to individual, living, or neighbourhood circumstances. To achieve this, we undertook an exploratory cross-sectional analysis of secondary data from a county-wide resident survey undertaken by the Cornwall Council in 2017.

## 3. Materials and Methods

### 3.1. Context

Cornwall, South West England, is distinct from other areas of the UK because it is largely rural in nature with dispersed settlement patterns. Over 40% of the local population live in settlements of less than 3000 people and around 573,299 individuals [34] reside in 230,400 households across Cornwall [35]. The area experiences pockets of high deprivation with 17 neighbourhoods falling in the top 10% of the most deprived areas in England in 2019 [36]. Moreover, fairly unique to the UK, Cornwall is dominated by a strong maritime climate characterised by mild temperatures, strong winds, and wet winters [37].

### 3.2. Resident Survey

In 2017, the Cornwall Council undertook a survey of a sample of residents to gauge satisfaction with the services the council provides and the local area, and assess quality of life [38]. Similar surveys were conducted in 2014, 2016, and on a rolling 6-month basis since 2018 [39]. In order to support further devolution and more local decision making across Cornwall, in 2015, Cornwall was split into 19 community network areas [40]. The survey responses were collected and assessed by the Community Network Area (CNA).

Unlike the other surveys, the 2017 survey included the short Warwick-Edinburgh Mental Wellbeing Scale (SWEMWBS [41,42,43]), which is used to assess changes in mental wellbeing, making the current study possible. The survey was sent out in the post to a random sample of at least 1250 households per CNA during the week commencing 3 July 2017, with reminders being sent 3–4 weeks later [38]. The closing date for responses was 21 August 2017, and of the 30,152 households who were sent the survey, 11,247 valid responses were received (38% response rate). There was the opportunity to respond online as well as by post, but only 4% of the responses came through the online option [38].

Any member of the household receiving the survey aged over 16 years old was eligible to complete the survey. The survey comprised the following 10 sections and participants were informed that it would take around 20 min to complete:About your local area and Cornwall Council;Contacting the Council;Community Safety;Respect and consideration;Your health (which included SWEMWBS [41,42,43]);Your home;Helping out (volunteering);Any other comments;Council newsletter (an opportunity to sign up for this);About you (socio-demographics).

Subsequently, the survey provided an opportunity to explore the potential contribution of a number of individual, living, and neighbourhood circumstances to mental wellbeing by housing tenure.

This study was a collaboration between Cornwall Council and the European Centre for Environment and Human Health, University of Exeter. Ethical approval for this cross-sectional secondary data analysis study was granted by the Cornwall Council Research Governance Framework panel on 3 August 2018. The University of Exeter Medical Schools ethics committee chair confirmed on 2 October 2018 that this was sufficient approval for a study only using anonymised data.

### 3.3. Statistical Analysis

The survey was reviewed to identify questions about participants’ socio-demographics, health, and wellbeing, and their attitudes toward their homes and local areas. Of the 66 questions in the survey, 24 were considered relevant and used in this study, the information obtained is listed in Appendix A. The questions not included related mostly to experience and satisfaction with council services. Raw SWEMWBS scores were transformed in line with standard practice [42]. As an exploratory cross-sectional study, listwise deletion was applied with no imputation of missing data. The final sample size for analysis was therefore 4085 (36% of the respondents); Appendix B lists the quantity of missing data for each variable and highlights where the complete and missing sample differed statistically significantly. Of this sample, 79% were residents of privately owned properties, 13% were residents of private rental properties, and 9% were residents of social housing properties. Initially, descriptive analyses were undertaken to compare each variable of interest across the three tenure types. Subsequently, univariable analyses were undertaken to understand the association between each variable of interest and mental wellbeing.

The same community or neighbourhood can often include privately owned, private rental, and social housing properties in the UK. We therefore wanted to explore whether renting or owning within the same area meant that different factors determined wellbeing, or the tenure type meant that the same factors had greater or lesser impact. The smallest geographical unit provided within the anonymised dataset was the CNA. Null two-level multilevel models were tested, stratified by tenure type with residents nested within the 19 CNAs. However, as the CNA interclass correlation coefficient (ICC) for the privately owned and social housing property residents was negligible, at less than 0.00001%, and for residents of private rental properties, the ICC was 0.4%, it was decided that clustering by area was not appropriate in this study.

The models were therefore only stratified by tenure type to allow associations between different factors and mental wellbeing in each tenure type to be identified within the constraints of a fairly small sample size in this study. Subsequently, the final models were fixed effects models and, as an exploratory study intended to generate hypotheses and research questions, a stepwise approach to the analysis was taken. All analyses were undertaken in Stata [44] with variables removed from the model with a *p*-value more than 0.08 and added to the model with a *p*-value less than 0.06. In line with the use of stepwise regression, the final models are reported without *p*-values.

## 4. Results

Responses to the survey questions of interest to this study, for the whole sample and by tenure, are listed in Appendix A, with the individual circumstance variables numbered IC1–IC13, the living circumstance variables numbered LC0–LC3, and the neighbourhood circumstance variables numbered NC1–NC41. Responses about many of the circumstances varied statistically significantly by tenure type (indicated in bold in Appendix A). Residents of private rental properties tended to be younger (IC1) than residents of privately owned properties, while the proportion of respondents who were female (IC2) or from an ethnic minority (IC3) was higher among residents of private rental or social housing properties than privately owned properties. The proportion of people living in more economically deprived areas (NC1) increased from privately owned to private rentals to social housing residents. On average, mental wellbeing (IC4) was highest among the residents of privately owned properties and the lowest among the social housing residents. Self-reported health (IC5), life satisfaction (IC8), having day-to-day activities limited by disability or illness (IC6), satisfaction with one’s home (LC1), and the ability to pay fuel bills (LC3) were the poorest among residents of social housing.

Patterns in responses to questions about living and neighbourhood circumstances were less consistent across housing type, with private rental residents sometimes reporting more dissatisfaction with the area in which they lived than social housing residents, despite more of the social housing residents living in areas of higher deprivation (NC1). Residents of privately rented properties were often more likely to report problems and issues in need of improvement than residents of privately owned or social housing properties, especially in relation to amenities in the local area. This may reflect the younger age profile of those living in private rental properties and the interests of those in or seeking work (e.g., salaries and transport). Figure 1 shows anonymised data on SWEMWBS scores by tenure within each of the 19 CNAs in Cornwall from the CNA with the lowest mean SWEMWBS score to the highest (left to right). Although the null multilevel models did not support modelling clustering by CNA, Figure 1 demonstrates that in 13 of the 19 CNAs, social housing residents reported lower mental wellbeing than residents of privately owned or rented properties.

The unadjusted associations between each of the individual, living, and neighbourhood circumstances and SWEMWBS scores (IC4) are reported in Appendix C. Many of the individual, living, and neighbourhood circumstances were separately found to be significantly associated with mental wellbeing, both before and after stratifying by tenure. However, the differences between the significant factors in Appendix A and Appendix C illustrate that greater reports of an issue do not necessarily equate to greater impact of that circumstance on mental wellbeing. Living circumstances, such as fuel poverty, and neighbourhood circumstances, such as safety and belonging, were found to be statistically significantly associated with wellbeing in the univariable analysis.

The marked differences in the number of residents of privately owned, private rental, and social housing properties, and the use of stepwise methods, meant that the number of variables retained in the models adapted to the sample size of each model. Subsequently, in the model of the 3218 respondents from privately owned homes, 17 variables remained in the model, whereas only 10 remained in the model of the 518 responses from those living in private rental accommodations and only 9 in the model of 349 responses from social housing residents. Health (IC5), social contact (IC7), and life satisfaction (IC8) were the only variables to be included in the models for every tenure type. The other variables remaining in each of the three tenure models may reflect distinct experiences and determinants of mental wellbeing (Appendix D).

Living circumstances, such as dissatisfaction with housing (LC1) and experiencing fuel poverty (LC3), were only found to be associated with poorer mental wellbeing among residents of privately owned and rented properties [45]. Those who were privately renting and reported problems with neighbourhood conditions, such as homelessness (NC32) and dogs in the area (NC39), or who felt the local sport and leisure facilities needed improving (NC22), reported poorer mental wellbeing. For those in privately owned properties, poor access to childcare (NC4) or shopping facilities (NC21), as well as problems with congestion (NC25), were associated with poorer mental wellbeing. While concerns about wages, the cost of living (NC26), and access to shopping facilities (NC21) were associated with lower mental wellbeing among social housing residents.

Residents of privately owned and social housing properties who felt safe in their local areas reported better mental wellbeing (NC28 and NC29). Concerns about a lack of respect and consideration in the local community (NC41) and perceptions of lower community integration (NC40) were only found to be associated with poorer mental wellbeing among residents of privately owned properties. Unexpectedly, residents of privately owned properties who reported more problems with loud music (NC36), and social housing residents who reported public drunkenness problems (NC30) reported higher mental wellbeing. Similarly, residents of privately owned or social housing properties who reported lack of satisfaction with the handling of anti-social behaviour and crime by the police and council (NC27) also reported higher mental wellbeing.

Together, the included variables explained around 50% of the variance in mental wellbeing in each model (adjusted R^2^ values). Figure 2 uses the Dahlgren and Whitehead model of the social determinants of health [2] and illustrates the different individual, living, and neighbourhood circumstances found to be associated with the mental wellbeing of residents of privately owned, privately rented, and social housing residents.

## 5. Discussion

Through secondary analysis of a 2017 resident survey in Cornwall, UK, it has been possible to generate some hypotheses about how individual, living, and neighbourhood circumstances contribute to inequalities in mental wellbeing related to housing tenure. Through simultaneously modelling individual, living, and neighbourhood circumstances we were able to explain around half of the variations in the mental wellbeing of residents in privately owned, private rental, and social housing properties. Better health (IC5) and satisfaction with life (IC8) were individual circumstances found to be associated with higher mental wellbeing across housing tenure sectors. The only other factor found to be associated with mental wellbeing across tenure types was social isolation (IC7), which was associated with worse mental wellbeing. Dissatisfaction with housing and fuel poverty (living circumstances) were only associated with poorer mental wellbeing among residents of privately owned and rented properties. Whereas concerns about safety in the local area (neighbourhood circumstances) were associated with mental wellbeing among residents of privately owned and social housing properties. Differences were found in the facilities and nuisances in the area associated with mental wellbeing across the tenure types, with some counterintuitive associations identified.

The counterintuitive findings may be an artifact of the stepwise regression approach employed. Tonn et al. [14] also found some less intuitive associations between wellbeing and housing in the USA, with those living in mobile homes or multifamily buildings experiencing less hardship than those in single family homes, which may indicate that neighbourhood circumstances, not just living circumstances, are important. In the current study, reporting issues within your community may reflect a degree of investment in the community, with these respondents feeling a degree of belonging to their neighbourhood and wider community. This hypothesis might be supported by the absence of these unusual associations among residents of private rental properties who may be more temporary in their current location than residents of privately owned or social housing properties. Residents of private rental properties reported more issues with amenities in their local area than community circumstances, such as safety and cohesion. The finding that residents of private rental properties who had caring responsibilities towards a family member, friend, or neighbour (IC10), amounting to 1–19 h per week, also had higher mental wellbeing, might reflect the greater sense of belonging for those individuals. Li et al. [13] found in Australia that the mental health of residents of private rental properties improved with stability in the rental situation and overtime equalised with that of residents of privately owned properties.

As with previous studies, living circumstances were found to be associated with mental wellbeing in the present study [11,17,46,47]. However, while fuel poverty and housing conditions are recognised as significant determinants of health, their impact on mental wellbeing was found to be limited to residents of privately owned and rented properties [45]. It is possible that private rental landlords may be less likely to invest in the property [28,48], which may explain this differential impact on wellbeing. Despite social housing residents having lower incomes and being more likely to have some health problems, this tenure group has generally attracted more investment, to raise the standard of housing and make the homes more affordable to heat. However, these interventions have resulted in variable health outcomes, with some resulting in adverse impacts on mental health, respiratory, and cardiovascular outcomes [28,45,48]. This is thought to result from a combination of resident behaviours, awareness levels, and inadequate heating and ventilation, despite interventions to make homes more affordable. This raises the need for public health interventions to consider whole house approaches (i.e., the built and human environment) to avoid these potential unintended consequences and make housing interventions more sustainable [4]. Despite reduced housing costs, social housing residents may still benefit from income interventions, with those in the current study who reported issues with cost of living and wages (NC26) also reporting poorer wellbeing [49].

Issues around neighbourhood conditions, such as access to services, e.g., shops (NC21) and childcare (NC4), might reflect the dispersed and rural geography of Cornwall, suggesting that place-based interventions may also support community wellbeing [4]. The findings that community integration (NC40) and respect (NC41) were only associated with mental wellbeing among residents of privately owned properties might suggest that more fundamental attributes of the neighbourhood, such as safety (NC28 and NC29), need to be addressed as a priority. As McAneney et al. [46] concluded, different interventions might be needed for different populations, or the same intervention might have different effects in different populations.

The importance of understanding and targeting the large range of housing related risk factors highlighted in this study, has been highlighted by the diverse impact of the COVID-19 pandemic on mental health and wellbeing [6,50,51]. The COVID-19 pandemic [52,53] has had both an impact on those with existing mental health conditions and increased the number of adults with self-reported depression [54]. Social isolation is considered a determinant of wellbeing across all ages, and has been exacerbated by the pandemic [55,56]. We found that, regardless of tenure, those who reported insufficient social contact (IC7) had markedly poorer mental wellbeing, demonstrating the need for more public health interventions addressing social isolation and loneliness. There are a number of person-centred interventions that can be adopted, such as volunteering, to improve health outcomes; however, more robust evidence of effectiveness among those experiencing social isolation is needed [57,58]. Overall, our findings support ‘connecting with people’ as one of the ‘five ways to wellbeing’, as promoted by the NHS [59,60]. However, addressing issues, such as sense of safety, might have greater impact on mental wellbeing, especially for social housing residents. It may be possible that by increasing social contact, fears about safety may be reduced.

The picture of individual, living, and neighbourhood circumstances identified within this study as impacting mental wellbeing resonates with findings from the C2 Programme (https://www.c2connectingcommunities.co.uk/ (accessed on 19 March 2022)) regarding the complex nature of interactions that affect wellbeing. The C2 implementation framework creates the conditions for new relations to form between residents and service providers in very low-income neighbourhoods. Part of the process of generating these partnerships involves listening to what people think is good about their neighbourhood and what they identify as the barriers to wellbeing. Having family, friends, and a sense of community are always discussed as positive aspects, whereas, dog fouling, litter, lack of safe play areas for children, poor and unaffordable transport, and not feeling safe are the issues most frequently cited as problems that the community would like prioritised by local services [61]. Taking a place-based approach to addressing the wider determinants of health and inequalities may deliver population level changes [62]. Such interventions need to adopt asset-based approaches and develop place-based partnerships, to better understand the individual and area-level needs and co-produce future policies and practices [63,64,65]. Future studies utilising these approaches need to be larger and employ longitudinal or experimental designs, especially when one might want to use interactions to explore differential effects by tenure, for example, the Watcombe Housing Study [29,66].

The existence of the Cornwall Council resident survey has enabled a low-cost study to be undertaken to explore the common finding that residents of social housing and private rental properties report poorer wellbeing than residents of privately owned properties. However, the use of survey data limits the study to a cross-sectional analysis, meaning that only associations, rather than causal relationships, can be explored. Furthermore, the limited sample size of the study and the large number of potentially relevant variables limited the analyses that could be undertaken. Stepwise regression is a poorly regarded analytical approach as it is entirely data driven, rather than hypothesis driven. However, within the context of this study, it was deemed appropriate for narrowing down hypotheses around the potential causes of wellbeing inequality. The statistical significance of the variables in Appendix D have not been reported as a reflection of the data-driven nature of the stepwise approach taken. Merlo et al. [67] describe more rigorous statistical methods for examining neighbourhood effects on health; however, these required a larger sample size than was available. As a consequence of these limitations, the lack of statistically significant associations in this study should be considered an absence of evidence of an effect, rather than evidence of an absence of an effect.

## 6. Conclusions

This study supports the need for more sustainable public health interventions involving whole house housing improvements to address fuel poverty and housing quality within the privately owned and rented sectors. Future interventions should also include measures to improve social cohesion, which could be achieved through promoting the NHS Five Ways to Wellbeing alongside other improvements. To better inform policy and practice, further interdisciplinary research is needed to explore the issues around income, access and safety among social housing residents in order to improve mental wellbeing. 

From a public mental health and wellbeing perspective, we also need a greater understanding of the role of diverse interventions alongside community-level approaches to improve sense of safety and connectedness. These should consider taking a whole system place-based people/community centred approach to reducing inequalities and avoid the potential unintended consequences of some prior interventions. Transdisciplinary research employing longitudinal study designs and mixed methods is needed to test and generate additional hypotheses around the poorer mental wellbeing often reported by residents of social housing and private rental properties in high income countries. These should also consider how the COVID-19 pandemic has impacted local people and communities, especially in light of the long-term implications caused by diverse socioeconomic impacts and health risks associated with long-COVID.

## Figures and Tables

**Figure 1 ijerph-19-03816-f001:**
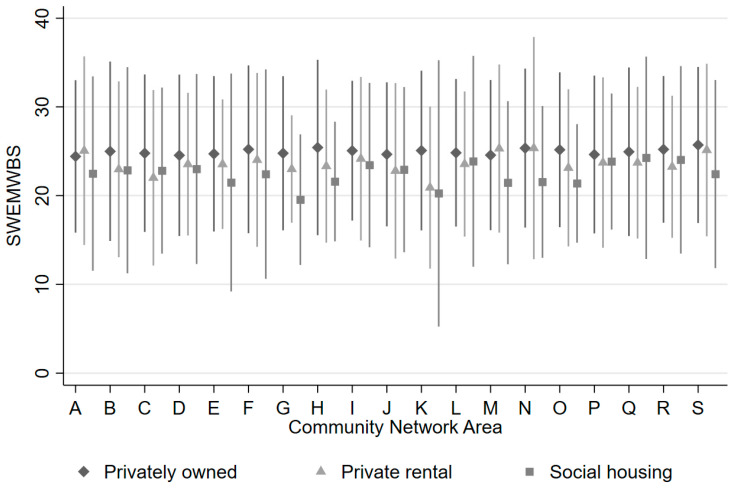
Mean (95% confidence interval) SWEMWBS score by tenure for each of the 19 Cornwall community network areas.

**Figure 2 ijerph-19-03816-f002:**
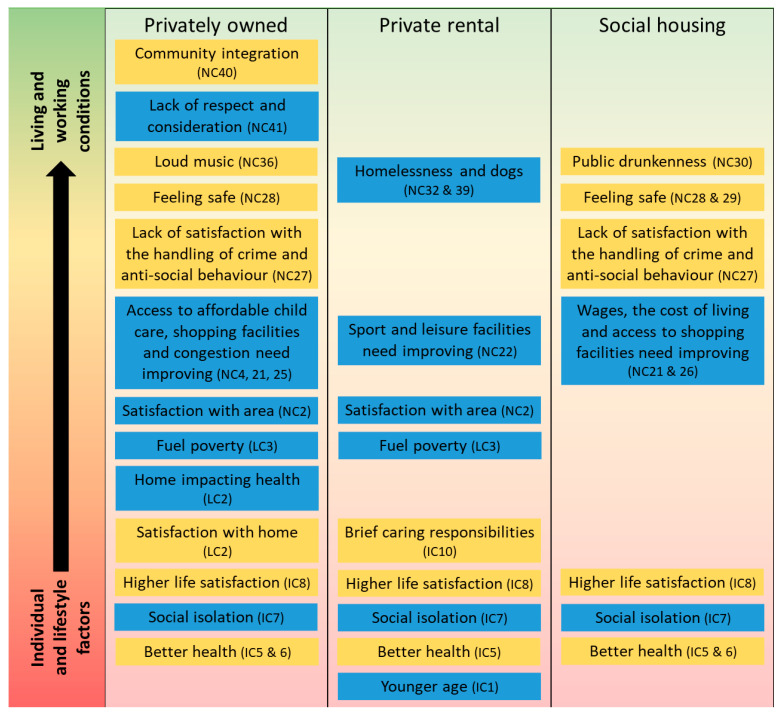
Individual, living, and neighbourhood circumstances contributing to mental wellbeing among homeowners, private renters, and social housing residents. Factors in yellow were associated with better mental wellbeing, while those in blue were associated with worse mental wellbeing.

## Data Availability

The data are not publicly available due to consent not being obtained for this at the time of data collection. The data are owned by Cornwall Council and the survey results are available here: https://old.cornwall.gov.uk/media/28979484/cornwall-residents-survey-full-report-2017.pdf (accessed on 19 March 2022).

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
