# Peer review of "Do the Determinants of Mental Wellbeing Vary by Housing Tenure Status? Secondary Analysis of a 2017 Cross-Sectional Residents Survey in Cornwall, South West England"

_ijerph, 2022, doi:10.3390/ijerph19073816_

Round 1

Reviewer 1 Report

Dear Authors,

Thank you for the opportunity to read this text. The article is interesting. The extensive regional research carried out in Cornwall is a good starting point. I think that the presented results of this research are the most valuable part of this article.

In my opinion, the text in its current version still needs to be refined before it is published. I have a few comments on the article, which will improve it.

The first issue concerns chapter 1. It has been named - Introduction. In my opinion, it is partly an Introduction and partly a Research Review. Unfortunately, including both of these elements in one chapter is not favorable. I recommend writing the Introduction and Research Review as two separate chapters.

In turn, in the chapter Review of research, I recommend referring to 3 or 4 discourses conducted in the world scientific literature concerning the thematic scope of this article. Of course, there are more such discourses. However, it would be expected to indicate three examples of discourses that, according to the Authors, fit in with their research.

I am surprised by the modest description of the research results. Extensive regional studies have been carried out, and the authors describe their results in 1.5 pages. Please expand the information about research results.

I think that the chapter Conclusions and References also needs strengthening.

Reviewer 2 Report

REPORT

The study takes into consideration an extremely current and interesting topic concerning housing conditions and their impact on mental health.

The conditions of daily living are one of the main causes of health inequalities. In this context, the home plays a fundamental role, considering the average time that the population spends there. To date, housing is one of the most important social determinants of health. In this context, housing adequacy means privacy, sufficient space, physical accessibility, environmental quality of housing and also structural stability and durability, and again, perceived safety, easy access to work and basic structures, and all at sustainable prices.

The study is clearly organized, contains all the components of a scientific article (introduction, Materials and methods, results, discussions, and conclusions); it needs to be integrated in some points (listed below), in the introduction section and in the discussion section.

The starting scenario of the study is clear, well detailed by the bibliographic references (to be integrated with further studies to investigate other environmental determinants in more detail), and the objective of the article is well defined. The method and the results are described in detail and are easily understood. Some additional considerations should, in my opinion, be integrated into the "Discussions" sections (as reported below).

Reading the document was, in my opinion, formative and stimulating, and can interest readers from different areas because the topic lends itself to being studied transversally, involving different disciplines.

The topic is topical and may be suitable for publication after minor revisions

 In particular

Introduction:

The section relating to the introduction is well organized and the information reported can be found in the literature. The particular theme being studied is, to date, one of the most debated in the literature, and the vision and general framework could be broadened by reporting further reflections and bibliographical references. To represent an exhaustive framework of the problem, it would be interesting, in my opinion, to describe in greater detail other factors as well, such as, for example: the housing context (neighborhood etc.) and how its conformation has an impact on the housing quality and mental health of the inhabitants; the physical characteristics of the accommodation (e.g., size, structure, conformation, indoor air quality, adequate environmental conditions, etc). All adequately supported by the literature evidence.

The introduction proposed by the authors is well organized and rightly referenced but it would be interesting to present the study considering most of the environmental factors involved. Tracing a much broader and more detailed overview of the proposed theme could better highlight and frame the complexity and specificity of the study.

Materials and method:

The section is clearly described, and all information is given in enough detail.

Results:

In the part of the questionnaire used in the study there are several questions concerning the surroundings of the house, the context, and more generally the neighborhood.

Obviously, the housing context has a great impact on living conditions, on the quality of housing and therefore on both physical and mental health. I would suggest to the authors to introduce some bibliographic references and some reflections concerning this topic both in the introduction and in the discussions.

Discussions:

In the section on discussions, the consequences of the pandemic on mental health in relation to living spaces are mentioned.

I would suggest more in-depth analysis and some additional reflection, perhaps reporting more recent studies that deal with a similar theme (both for the purpose of the study, both for the tools used, and for the topics covered) to that proposed in this study by the authors. It could be interesting to compare the results of the most recent investigations conducted downstream of the most difficult moment of the COVID 19 pandemic with the results of the proposed study precisely to corroborate the authors' conclusions and to update the study.

Conclusions:

In this section it would be important to also mention the need for an interdisciplinary approach in addressing this particular and very important topic.

Reviewer 3 Report

I had read the paper carefully, and found that everything was very well organized and written. It has a strong conceptual framework; and analysis as well as the discussion are clear. The only concern might be using the secondary data (or a little old data) which seems somehow acceptable in the uncertain era of the pandemic that we are living in and makes data collection really difficult. But, in general I agree with the overall manuscript in its presence format.   

Round 2

Reviewer 1 Report

Dear Authors,

After the corrections, the article is more readable. I only have two more remarks. I think the research's aim should be emphasized in the Introduction, and the research gap should be indicated. Tables 1 and 2 are pretty extensive, and maybe it might be better to move them to an Appendix.
